# Early to Late VSV-G Expression in AcMNPV BV Enhances Transduction in Mammalian Cells but Does Not Affect Virion Yield in Insect Cells

**DOI:** 10.3390/vaccines13070693

**Published:** 2025-06-26

**Authors:** Jorge Alejandro Simonin, Franco Uriel Cuccovia Warlet, María del Rosario Bauzá, María del Pilar Plastine, Victoria Alfonso, Fernanda Daniela Olea, Carolina Susana Cerrudo, Mariano Nicolás Belaich

**Affiliations:** 1Laboratorio de Ingeniería Genética y Biología Celular y Molecular (LIGBCM), Instituto de Microbiología Básica y Aplicada, Comisión de Investigaciones Científicas de la Provincia de Buenos Aires, Universidad Nacional de Quilmes, Buenos Aires B1876BXD, Argentina; jorge.simonin@unq.edu.ar (J.A.S.); franco.cuccovia@unq.edu.ar (F.U.C.W.); ccerrudo@unq.edu.ar (C.S.C.); 2Laboratorio de Medicina Regenerativa Cardiovascular, Instituto de Medicina Traslacional, Trasplante y Bioingeniería (IMETTYB), Consejo Nacional de Investigaciones Científicas y Técnicas (CONICET), Universidad Favaloro, Buenos Aires C1078AAI, Argentina; mbauza@favaloro.edu.ar (M.d.R.B.); dolea@favaloro.edu.ar (F.D.O.); 3Instituto de Agrobiotecnología y Biología Molecular (IABIMO), Instituto Nacional de Tecnología Agropecuaria (INTA), Consejo Nacional de Investigaciones Científicas y Técnicas (CONICET), Hurlingham, Buenos Aires B1686WAA, Argentina; plastine.maria@inta.gob.ar

**Keywords:** baculovirus, AcMNPV, VSV-G, viral-based gene delivery

## Abstract

**Background/Objectives:** Baculoviruses represent promising gene delivery vectors for mammalian systems, combining high safety profiles with substantial cargo capacity. While pseudotyping with vesicular stomatitis virus G-protein (VSV-G) enhances transduction efficiency, optimal expression strategies during the Autographa californica multiple nucleopolyhedrovirus (AcMNPV) infection cycle remain unexplored. This study investigates how VSV-G expression timing affects pseudotype incorporation into budded virions (BVs) and subsequent transduction efficacy. **Methods:** Three recombinant AcMNPV constructs were generated, each expressing VSV-G under distinct baculoviral promoters (*ie1*, *gp64*, and *p10*) and GFP via a CMV promoter. VSV-G incorporation was verified by Western blot, while transduction efficiency was quantified in mammalian cell lines (fluorescence microscopy/flow cytometry) and rat hind limbs. Viral productivity was assessed through production kinetics and plaque assays. **Results:** All the pseudotyped viruses showed significantly enhanced transduction capacity versus controls, strongly correlating with VSV-G incorporation levels. The *p10* promoter drove the highest VSV-G expression and transduction efficiency. Crucially, BV production yields and infectivity remained unaffected by VSV-G expression timing. The in vivo results mirrored the cell culture findings, with p10-driven constructs showing greater GFP expression at low doses (10^4^ virions). **Conclusions:** Strategic VSV-G expression via very late promoters (particularly *p10*) maximizes baculoviral transduction without compromising production yields. This study establishes a framework for optimizing pseudotyped BV systems, demonstrating that late-phase glycoprotein expression balances high mammalian transduction with preserved insect-cell productivity—a critical advancement for vaccine vector development.

## 1. Introduction

Baculoviruses are insect pathogens that possess double-stranded DNA genomes, ranging from 80 to 180 kbp, which are infective per se in susceptible hosts and packed within a rod-shaped nucleocapsid, hence the name “Baculo”-virus. These nucleocapsids are enveloped by two different membranes according to the time of infection, generating the Occluded Derived Virions (ODVs, immersed within protein crystals called Occlusion Bodies—OBs) and the budded virions (BVs), the former being responsible for the primary infection in nature and the latter (in the baculoviral species that generate them) being those that spread the infection within the affected individual. Typically, baculoviruses possess a narrow host range, which makes them valuable tools as biopesticides [1,2]. Since it was discovered that the BVs of baculoviruses can enter mammalian cells without initiating a replicative cycle, they have been proposed as alternative viral vectors for gene therapy and viral vector-based vaccines [3,4,5], with *Alphabaculovirus aucalifornicae* (Autographa californica multiple nucleopolyhedrovirus, AcMNPV) being the most extensively utilized species [6]. A significant amount of research over the last two decades has shown their potential.

The baculovirus infection process is temporally regulated and is usually divided into three main phases: early (sub-divided into immediate early and delayed early); late; and very late [7]. These phases are defined by the hours post-infection (hpi) and the different viral genes being expressed. The early phase starts upon attachment, entry, endosomal escape, nuclear entry, uncoating, viral gene expression, and the subsequent halting of host transcription. Early viral genes are transcribed by the host cell RNA polymerase II and also utilize host transcription factors; this occurs within 0.5 to 6 hpi [8,9], although some essential gene promoters remain active during the late stages of the infection cycle [10,11].

The constitutive *immediate early 1* promoter is widely used in baculovirus expression systems due to its activity throughout all infection phases and in uninfected insect cells [12,13]. At ~6 hpi, viral DNA replication triggers late gene transcription [8], marked by TAAG motif recognition by the viral RNA polymerase, which ensures viral gene expression dominance over host machinery [14,15]. Late genes encode structural proteins (e.g., BV glycoprotein GP64 in *Alphabaculovirus*) essential for BV production (12–18 hpi) and ODV assembly [16].

The very late phase (18 hpi to lysis) is characterized by ODV formation and the hyperexpression of *polyhedrin* and *p10* [17]. Transcriptional bursts during this phase maximize viral protein production as host gene expression declines [1]. Very late promoters (starting at 18–24 hpi) surpass all the others in activity [18]; for instance, *p10* (involved in OB maturation and lysis) activates earlier than *polyhedrin* but at slightly lower levels [19].

In contrast with other viral vectors, baculoviruses possess several qualities that make them appealing as gene therapy vectors and viral vector-based vaccines. They possess a level 1 biosafety profile, as they are not mammalian viruses and do not replicate in them. Moreover, thanks to the availability of numerous commercial platforms and cell lines, recombinant baculovirus stocks can be obtained quickly and easily and produced at relatively high titers. Furthermore, they can accommodate large DNA sequences in their genomes and do not integrate into host cells, enhancing their versatility and safety profile as gene delivery vectors [20,21]. Due to these advantages, baculoviruses have already been used in several successful studies, including vaccines [20,22], cancer treatment [23,24,25,26,27,28,29,30], cardiac disease treatments [31,32,33,34,35], and regenerative medicine [36,37,38,39,40,41,42], demonstrating their versatility. While their transient gene expression (7–14 days) [43] may limit certain applications, this feature is particularly advantageous for genome editing strategies where controlled, short-term expression is desirable for safety. Other limitations include variable transduction efficiency that depends on the target cell type. Although baculoviruses possess broad tissue and host tropism due to their ability to enter both quiescent and proliferating cells, their entry efficiency (including in this aspect the correct release from endosomes) can sometimes be a limitation, hindering the potency of the vector and requiring the utilization of high quantities of the virus.

Although the transduction mechanism of AcMNPV is still unclear and requires more research, the relevance of the GP64 glycoprotein in mediating viral entry is well-established [44]. This factor is crucial for viral attachment, internalization, and endosomal membrane fusion during the transduction process [45]. Baculovirus entry into mammalian cells varies by cell type, with clathrin-mediated, low pH-dependent endocytosis being predominant. However, fusion in early endosomes often limits transduction efficiency [46]. Alternative pathways have been reported, including caveolae-mediated entry (supported by genistein-induced transduction enhancement in BHK21 cells [47]) and phagocytosis-like mechanisms in hepatocytes, independent of clathrin or macropinocytosis [48]. Pseudotyping with heterologous glycoproteins can overcome these barriers. The VSV-G protein, widely used in lentiviral and retroviral systems [49], has been successfully adapted for AcMNPV, improving both host range and endosomal escape. This strategy enhances the nuclear delivery of nucleocapsids [50,51,52,53,54], addressing a key bottleneck in transduction efficiency. In this study, our objective was to assess the effect of the promoter temporality chosen for the VSV-G expression on its incorporation by the BV and how this translated to an improved transduction efficiency and affected infective capacity in insects. For this purpose, baculoviral promoters representative of each phase of the infection cycle (promoters of the *ie-1*, *gp64*, and *p10* genes) were evaluated to express VSV-G. While late promoters (e.g., *p10* and *polyhedrin*) drive higher expression levels, earlier promoters (e.g., *ie1* and *gp64*) may offer advantages for pseudotyping AcMNPV with VSV-G: their activity during earlier infection stages ensures prolonged protein accumulation in a less compromised cellular environment, potentially enhancing glycoprotein functionality and viral progeny quality. Previous studies evaluating AcMNPV promoters for recombinant protein production (including cytoplasmic and secreted proteins) demonstrated that earlier, weaker promoters often yield higher protein quantities and/or superior quality compared to the hyperactive *polyhedrin* promoter, particularly for proteins requiring complex post-translational modifications or efficient secretion. These findings directly informed our selection of promoters to evaluate for the VSV-G pseudotyping of AcMNPV BVs [55].

## 2. Materials and Methods

### 2.1. Bioinformatics Studies

The VSV-G and GP64 proteins were analyzed for sequence and structural features. Amino acid sequences were retrieved from the UniProtKB/SwissProt database (https://www.uniprot.org/ accessed on 15 April 2025). Protein motifs, patterns, and domains were identified using InterPro (https://www.ebi.ac.uk/interpro/search/sequence/ accessed on 15 April 2025) and PROSITE (https://prosite.expasy.org/ accessed on 15 April 2025) [56,57]. Secondary structure predictions were generated with JPRED4 (https://www.compbio.dundee.ac.uk/jpred/ accessed on 15 April 2025) [58], and consensus structures were derived by integrating DSSP-based secondary structure assignments from the proteins’ three-dimensional structures. Signal peptides were predicted using SignalP-6.0 (https://services.healthtech.dtu.dk/services/SignalP-6.0/ accessed on 15 April 2025) [59]. Default parameters were applied for all tools. Moreover, hydrophobicity profiles were calculated with the ProtScale server (https://web.expasy.org/protscale/ accessed on 15 April 2025) using the Kyte–Doolittle scale and a 21-amino-acid sliding window [60]. For tertiary structure analysis, the atomic coordinates of GP64 (PDB: 3DUZ) and VSV-G (PDB: 5I2M) were obtained from the RCSB Protein Data Bank (https://www.rcsb.org/ accessed on 15 April 2025). Structural relationships were assessed via pairwise alignments and superpositions using the Dali server (http://ekhidna2.biocenter.helsinki.fi/dali/ accessed on 15 April 2025) [61], followed by manual refinement. Molecular graphics were rendered with UCSF ChimeraX [62].

### 2.2. Maintenance of Insect and Mammalian Cell Lines

The insect cell line Sf9, derived from *Spodoptera frugiperda* [63], was used for the generation and multiplication of AcMNPV recombinant viruses derived from the Bac-to-Bac^®^ Baculovirus Expression System (Thermo Fisher Scientific, Waltham, MA, USA). In all cases, the cells were maintained at 27 °C (MIR 5531 oven, SANYO, Kadoma, Osaka, Japan) in a monolayer in polystyrene flasks of 25, 75, and 175 cm^2^ using Grace’s insect medium (GRACE’s, pH 6.18; Thermo Fisher Scientific), supplemented with 10% *v*/*v* fetal bovine serum (FBS; Natocor, Córdoba, Argentina) and antibiotics and antimycotic (Thermo Fisher Scientific). Meanwhile, the mammalian cell lines used in this work were maintained at 37 °C in a monolayer in 25 and 75 cm^2^ polystyrene flasks using Dulbecco’s Modified Eagle Medium (DMEM, Gibco, Grand Island, NY, USA), with 10% inactivated FBS plus antibiotics and antifungals (Thermo Fisher Scientific), in a humidified atmosphere with 5% CO_2_. All the cell lines were used at passages between 3 and 10 for the assays conducted in this study.

### 2.3. Generation of Recombinant Plasmids

Recombinant BVs encoding the green fluorescent protein (GFP) and the glycoprotein (G protein) of vesicular stomatitis virus (VSV-G) under different baculovirus promoters were generated using the bacmid bMON14272 from the Bac-to-Bac^®^ Baculovirus Expression System (Thermo Fisher Scientific) [64] according to the manufacturer’s recommendations. The pFastBac-Dual (Thermo Fisher Scientific) was used as a backbone to generate the plasmids used to create the recombinant baculoviral genomes by transposition according to the Bac-to-Bac^®^ system. All PCR reactions were carried out using Pfu or Taq DNA polymerase (PB-L, Buenos Aires, Argentina) under standard thermal conditions, and DNA transformations in bacteria were performed by electroporation in *Escherichia coli* Top 10. The promoters of the AcMNPV *ie1* (662 bp upstream of the start codon) and *gp64* (272 bp upstream of the start codon) genes were obtained by PCR using the primers Fw.pIE1/Rv.pIE1 and Fw.pGP64/Rv.pGP64 (Table 1), respectively, with the AcMNPV viral genome (GenBank: L22858.1) as the template. The amplicons obtained were ligated to the TOPO-type plasmid pEasy-T1 (TransGen Biotech, Beijing, China), resulting in the pEasy-IE1 and pEasy-GP64 constructs. These plasmids were then digested with HindIII and BamHI to allow the subsequent subcloning of the GFP ORF derived from pFastBac-CMV-GFP [34] treated with HindIII and BglII. Later, the resulting constructs were digested with HindIII and XhoI and ligated to the pFastBac-Dual linearized with the same enzymes, generating the constructs pFastBacPromIE1/GFP and pFastBacPromGP64/GFP (Table 2). In the case of the *p10* promoter, molecular cloning was not required, as a construct compatible with the objectives of the work was previously available [65], named in this work as pFastBacPromp10/GFP and carrying the AcMNPV *p10* promoter already present in pFastBac-Dual (Table 2). On the other hand, the VSV-G ORF was recovered by PCR (1567 bp) using the primers FwVSVG/RvVSVG (Table 1) and the plasmid pCAG-VSVG (Addgene #35616) as a template. The amplicon obtained was ligated to pFastBac-CMV-GFP, previously linearized with PvuII, to generate pFastBacPromP10/VSVG (Table 2). This plasmid was then digested with BamHI to eliminate the *p10* promoter and to subclone the *ie1* and *gp64* promoters derived from pEasy-IE1 and pEasy-GP64, treated with the same enzyme, generating pFastBacPromIE1/VSVG and pFastBacPromGP64/VSVG (Table 2). In all the cases, the generated constructs were verified by physical maps and Sanger sequencing (Macrogen, Seoul, Repiblic of Korea).

### 2.4. Generation of Recombinant Baculoviruses

The recombinant baculoviral genomes obtained from transposed DH10Bac bacteria, previously transformed with the different versions of modified pFastBac-Dual, were transfected into the monolayers of Sf9 cells grown in Grace’s medium (Thermo Fisher Scientific) supplemented with 10% *v*/*v* FBS (Natocor) using the polyethylenimine (PEI) 87 kDa reagent and reported procedures [66]. The viral progeny obtained (BV.IE1-GFP, BV.GP64-GFP, BV.P10-GFP, BV.IE1-VSVG, BV.GP64-VSVG, BV.P10-VSVG, and BV.CMV-GFP; Table 2) were recovered from the conditioned media after 7 days and used as starter stock to propagate the BV through infection assays at low multiplicities to avoid the generation of defective virions. The propagation, titration, and storage of the baculovirus BV stocks were carried out as previously reported [67,68,69]. The viral batches produced were correctly identified and stored at 4 °C, protected from light. Subsequently, the multiplication of BVs was carried out by infecting the cultures of Sf9 cells grown in monolayers in 175 cm^2^ flasks at 70% confluency using multiplicities of infection (MOIs) of less than 1, and incubating for 4 days at 27 °C. The clarified supernatants of the infected Sf9 cells were centrifuged over a 30% *w*/*v* sucrose cushion at 22,000 rpm for 2 h and 30 min at 4 °C (Beckman XL-70 ultracentrifuge, SW28 rotor; Beckman Coulter, Brea, CA, USA) and resuspended in PBS pH 7.4 buffer. The concentrated viruses were used for in vitro and in vivo transduction assays and for performing Western blots. The viral stocks were titrated in terms of their capacity to form infectious foci (Foci-Forming Unit per mL; FFU/mL) according to the reported methods [70] and using fluorescence microscopy (Cytation 5, Biotek Instruments, Winooski, VT, USA).

### 2.5. Expression Timing Assay of the ie-1, gp64, and p10 Promoters

Sf9 cells grown in 6-well plates at 70% confluence were infected at an MOI of 5 with BV.IE1-GFP, BV.GP64-GFP, and BV.P10-GFP (Table 2) according to the reported methods [70]. Sf9 cells without infection were used as a negative control. Post-infection, the cells were extensively washed with PBS (four times) to remove any BV from the original inoculum that did not enter the cells and were then maintained in growth conditions with 2 mL of complete medium. Evidence of green fluorescence was analyzed by microscopy (Cytation 5, Biotek) every 3 h for 48 h. Additionally, 50 µL samples of conditioned medium were harvested at the same time intervals and subsequently used as infection inocula on Sf9 cells growing in 96-well plates. After 10 days of incubation, the presence or absence of BV in the original samples (progeny virions resulting from infection) was analyzed by determining green fluorescence.

### 2.6. VSV-G Immunodetection by Western Blot

To perform the Western blots, the virus samples (4.5–9 × 10^7^ FFU/mL) of BV.IE1-GFP, BV.GP64-GFP, and BV.P10-GFP (prepared as indicated in Section 2.4) were separated by SDS-PAGE (10% acrylamide) and transferred to a polyvinylidene fluoride membrane (PVDF, GE Healthcare, Chicago, IL, USA) using the Trans-Blot^®^ SD semi-dry transfer cell (BioRad, Hercules, CA, USA). After transfer, the membranes were blocked by incubation overnight at 4 °C in PBS buffer (1.8 mM Na_2_HPO_4_, 10 mM KH_2_PO_4_, 137 mM NaCl, 2.7 mM KCl; pH 7.4) with 5% *w*/*v* bovine serum albumin (BSA). For the incubation of the primary antibodies (Mouse anti-VSV-G Antibody and Mouse anti-GP39 Antibody, Santa Cruz Biotechnology, Dallas, TX, USA), dilutions between 1:500 and 1:1000 were used in PBS buffer with 1% *w*/*v* BSA and 0.1% *v*/*v* Tween-20 at 37 °C for 1 h. Then, for the incubation of the secondary antibody (Goat anti-mouse IgG-HRP, Santa Cruz Biotechnology), 1:10,000 dilutions in PBS buffer with 0.1% Tween-20 were used at 37 °C for 1 h. After each antibody incubation, the membrane was washed three times with PBS buffer containing 0.1% Tween-20 for 5 min at room temperature. The development of the membranes was carried out in a dark room using radiographic plates and Bio-Lumina reagent (BioRad). The band intensities were quantified by a densitometric analysis using ImageJ 1.54g [71]. VSV-G incorporation levels were normalized to VP39, and the relative VSV-G incorporation in each recombinant virus was then compared against that detected in BV.IE1-GFP.

### 2.7. Pseudotyped BV Infection Kinetics in Insect Cells

Sf9 cells grown in 6-well plates at 70% confluence were infected at an MOI of 5 with the BV.IE1-VSVG, BV.GP64-VSVG, and BV.P10-VSVG viruses, and the BV.CMV-GFP virus (Table 2) was used as a negative control (in triplicate). After the infection process, the cells were washed 4 times with 1 mL of PBS before adding 1 mL of complete Grace’s medium. Post-infection, the supernatant from the infected cells was collected every 6 h for 48 h. Each time the supernatant was taken, the cells were washed twice with PBS to remove any remaining viruses, and 1 mL of complete Grace’s medium was added. The recovered supernatants were titrated using standard methods to measure their capacity to form infectious foci (FFU/mL) [70]. Additionally, Sf9 cells grown in 24-well plates at 80% confluence were infected with dilutions of the viral stocks to achieve approximately 1–5 foci of infection per well. After infection, all the medium was removed, and 500 μL of liquid low-melting agarose (3% *w*/*v*) mixed at 50% with complete Grace’s medium was added. Once the agarose had solidified, complete medium was added, and the cells were incubated at 27 °C. The progression of the size of the infection foci was monitored by fluorescence microscopy (Cytation 5, Biotek) every 24 h. The horizontal and vertical diameters of each infection focus were measured. The results were analyzed by two-way ANOVA followed by Dunnet’s post-test. The GraphPad Prism 10 software (GraphPad Software Inc., San Diego, CA, USA) was used. Statistical significance was set at *p* < 0.05. The results were expressed as mean ± standard deviation.

### 2.8. Syncytium Formation Assay

Sf9 cells grown in 6-well plates at 100% confluence were infected at an MOI of 5 with the BV.IE1-VSVG, BV.GP64-VSVG, and BV.P10-VSVG viruses, and the BV.CMV-GFP virus (Table 2) was used as a negative control (in triplicate). Seventy-two hours post-infection, the cells were fixed with 4% *v*/*v* formaldehyde in PBS and stained with 4′,6-diamidino-2-phenylindole (DAPI). Syncytium formation was analyzed by fluorescence microscopy (Cytation 5, Biotek).

### 2.9. Mammalian Cell Lines Transduction Assay

The mammalian cell lines evaluated (HEK 293T, Vero, and MIA-PaCa-2) were seeded in 6-well plates and maintained at 37 °C in DMEM medium with 10% *v*/*v* inactivated FBS, along with antibiotics and antimycotics, in a humid atmosphere with 5% *v*/*v* CO_2_, until reaching 70–80% confluence. For transduction with the recombinant viruses, the cells were washed with PBS (pH 7.4), and a 500 µL mixture of concentrated virus/PBS was added according to the MOI to be used, ensuring that the cells were covered with the minimum volume of liquid possible. The cells with the virus were then incubated at 37 °C for 1 h, after which the PBS with the virus was removed and complete DMEM medium was added. Transduction efficiencies were analyzed qualitatively by fluorescence microscopy (Cytation 5, Biotek) and quantitatively as the percentage of green-positive cells by flow cytometry (FACS Calibur, BD Biosciences, Franklin Lakes, NJ, USA) at 24 h post-transduction.

### 2.10. Hind Limb Rat Transduction Assay

Twenty-eight rats weighing between 300 and 350 g were anesthetized by the intraperitoneal injection of xylazine (10 mg/kg) and ketamine (50 mg/kg). Each animal received three injections of 100 µL into the hind limb, containing different doses of baculovirus according to group assignment: BV.CMV-GFP or BV.P10-VSVG (10^8^, 10^6^, or 10^4^ total viral copies; 4 replicas were performed per dose and group). Of the total 28 animals, 4 were used as controls and received three injections of 100 µL of PBS. After 3 days, the animals were euthanized, and tissue samples were collected for histological analysis. Cryosections were obtained using Cryoplast (Biopack, Buenos Aires, Argentina). The cryosections were stained with DAPI to label nuclei, or with hematoxylin and eosin (H&E) to evaluate tissue morphology. The samples were analyzed under a fluorescence or optical microscope. Images were acquired at 20× magnification, and in some cases at 40× to visualize skeletal muscle striations, using an Axio Observer inverted microscope (Zeiss, Oberkochen, Germany).

All the experiments were performed in accordance with the Guide for Care and Use of Laboratory Animals [72] and approved by the Laboratory Animal Care and Use Committee of the Favaloro University (approval CICUAL-UF 2025-002).

## 3. Results and Discussion

### 3.1. VSV-G and GP64 Comparative Studies

The bioinformatics analysis revealed structural similarity between VSV-G and GP64 despite their divergent amino acid sequences and evolutionary lineages. The fusion and trimerization domains emerged as the most conserved regions, with the fusion domain showing particularly high structural conservation (Figure 1A). While VSV-G’s trimerization domain is structurally defined, GP64’s is sequence-dependent (Figure 1B,C). Both proteins exhibited comparable hydrophobicity profiles, with peaks corresponding to their signal peptides and transmembrane regions in equivalent positions (Figure 1A). Structural alignment of the homotrimers demonstrated strong superposition, confirming the similarity of these functional domains (Figure 1D). To evaluate potential heterotrimer formation, we modeled biological assemblies containing mixed compositions of VSV-G and GP64 monomers (1GP64:2VSV-G and 2GP64:1VSV-G). These predictions revealed that configurations with two VSV-G monomers (1GP64:2VSV-G) were more likely to form, showing greater conservation in the fusion domain and more favorable interactions between trimerization domains (Figure 1E). Based on these structural insights, we hypothesize that recombinant AcMNPV expressing VSV-G at different infection timepoints will generate BVs with varying exogenous glycoprotein loads. While this heterogeneity might enhance mammalian transduction efficiency, it could potentially compromise virion productivity in insect cells. Collectively, these results suggest that increased VSV-G availability could enhance transduction efficiency through both VSV-G homotrimer formation and the generation of structurally compatible VSV-G/GP64 heterotrimers. These structural predictions, while informative, remain speculative and would require experimental validation through the direct examination of VSV-G and GP64 spatial arrangement in BV envelopes—an analysis beyond the scope of the current study, which primarily focuses on characterizing BV activity in insect and mammalian systems.

### 3.2. Recombinant Baculoviruses Expressing VSV-G

To generate BVs from the baculovirus AcMNPV pseudotyped with different amounts of VSV-G in their envelopes, the activity of three baculoviral promoters (selected from genes with expression profiles consistent with the three main phases of baculoviral gene expression) was first studied based on the kinetics of viral progeny production (in their BV phenotype) during a 48 h infection cycle in susceptible cells. In this regard, three recombinant baculoviruses named BV.IE1-GFP, BV.GP64-GFP, and BV.P10-GFP that express GFP under *ie1*, *gp64*, and *p10* promoters, respectively, were generated (Figure 2A) and analyzed in the Sf9 cells by fluorescence microscopy at different post-infection times to determine the onset of the different promoters and their relationship with the generation of BVs (Figure 2B). As expected (and considering that detectable GFP fluorescence appears later due to the combined kinetics of translation, chromophore oxidation, and folding), the immediate early promoter *ie1* showed activity as early as 6 hpi and remained active throughout the entire infection cycle, as it is recognized by the transcription factors and RNA polymerase II from the host [6,7]. The late *gp64* promoter became active at 12 hpi, and the very late *p10* promoter at 24 hpi, as previously described for both genes [14,18]. The kinetics of BV production overlapped with the GFP production time driven by the different promoters, showing that the majority of BV production occurs after 24 hpi when all three promoters are active (Figure 2C). This result allows us to hypothesize that any of the three promoters would be useful in driving the production of VSV-G pseudotyped virions.

Considering the above, three recombinant baculoviruses (BV.IE1-VSVG, BV.GP64-VSVG, and BV.P10-VSVG) were generated to express VSV-G at different times during the infective cycle in susceptible insect cells using the previously tested promoters. For this, the wild-type version of the glycoprotein was used, assuming that its signal peptide and transmembrane region would be adequate to direct polypeptide synthesis in the rough endoplasmic reticulum and enable anchoring in the cytoplasmic membrane, which would subsequently incorporate into the BV envelope. Once virions were produced, the presence of VSV-G in the BVs was verified by Western blot analysis using the nucleocapsid VP39 protein as a loading control (Figure 3 and Appendix A). In all the pseudotyped virions, VSV-G was detected, but in varying amounts. The BV.P10-VSVG showed the highest VSV-G incorporation compared to the others, indicating that strong expression of the glycoprotein at very late times in the infection cycle was more effective than sustaining a constant expression from the beginning of BV production. The generated recombinant viruses retain the gp64 gene. We consider this retention important, as its elimination during VSV-G pseudotyping can negatively impact insect cell productivity [51]. Notably, previous reports using the polyhedrin promoter to express the pseudotype have documented no apparent adverse effects under such conditions [50,51,52,53,54].

Moreover, as it has been reported that the production of VSV-G in Sf9 cells stimulates cell fusion in the slightly acidic pH conditions of insect cell culture medium [54], a syncytium formation assay was performed, demonstrating the activity of the fusogenic protein (Figure 4). Consistent with the Western blot results, where BV.P10-VSVG expressed the highest amount of glycoprotein, this virus stock produced a significant cell fusion effect in the insect cells, which was stronger than that caused by BV.GP64-VSVG and BV.IE1-VSVG.

### 3.3. Infectious Capacity in Insect Cells

It was previously reported that BVs expressing VSV-G, where the GP64 gene was deleted, were capable of infecting insect cells and generating BV progeny, but in lower quantities [51]. Therefore, it was of interest to test if the expression of the pseudotype was affecting the virus infectivity and the yield of virions being produced. In this regard, the BV production by the viruses carrying VSV-G was evaluated and compared to a non-pseudotyped virion, showing no significant differences in yield among them (Figure 5).

Simultaneously, we evaluated whether the infective power was affected. To this end, the size and propagation time of infection foci in the monolayers of the Sf9 cell cultures embedded in agarose were analyzed (Figure 6). Viral foci were identified and tracked every 24 h by fluorescence microscopy, and the vertical and horizontal diameters were measured, with no significant differences found among the tested virions. The combination of these two studies allowed us to confirm that the expression of VSV-G at different times during the AcMNPV infective cycle does not affect the production or infective capacity of the resulting BV.

### 3.4. Transduction Activity in Mammalian Cells

The transduction efficiency of the pseudotyped and non-pseudotyped recombinant BVs was compared by incubating the Vero, HEK 293T, and MIA PaCa-2 cells with the different baculoviruses at an MOI of 100 and analyzed by fluorescence microscopy and flow cytometry at 24 h post-transduction (Figure 7). In all the cell lines, the number of green-positive cells was higher in pseudotyped BV compared to the non-pseudotyped control. Particularly, BV.P10-VSVG was the one that obtained the most notable results, almost doubling the transduction efficiency of the control virus. Meanwhile, virions pseudotyped by the expression of VSV-G at early and late times of the infective cycle in insects also showed an improvement in their mammalian transducing capacity, although less pronounced. All these tests would show that the amount of VSV-G in the BV envelopes determines their capacity for improvement in mammals, and that to achieve BV with a high VSV-G load, its expression is required at very late times. The selected MOI aligns with established baculovirus transduction protocols for animal studies [20,34,35], though future dose–response experiments could provide valuable mechanistic insights into transduction efficiency thresholds.

For the best-performing pseudotyped virus (BV.P10-VSVG), we conducted an in vivo transduction assay by injecting varying quantities of infectious virus particles (IVPs) into rat hind limbs, using the non-pseudotyped virus (BV.CMV-GFP) as a control (Figure 8). Three days post-injection, the animals were euthanized humanely, and skeletal muscle tissue samples from the injection point were collected for histological analysis. Cryosections were stained with H&E to evaluate tissue morphology or with DAPI and examined by fluorescence microscopy.

At doses of 10^8^ and 10^6^ IVP, BV.P10-VSVG and BV.CMV-GFP exhibited comparable GFP expression levels at 3 days post-transduction. In contrast, at 10^4^ IVP, the GFP expression was markedly higher in the animals injected with the pseudotyped virus than in those receiving the non-pseudotyped control. These findings align with prior observations from cell culture transductions, and demonstrate that pseudotyping enables a 10,000-fold reduction in viral load while maintaining high coverage and efficient expression of the protein of interest in muscle tissue. This dose-sparing effect highlights the translational potential of BV.P10-VSVG for gene delivery applications.

## 4. Conclusions

In recent years, human medicines based on nucleic acids as active ingredients have emerged with great success, enabling diagnostic, preventive, or therapeutic services through gene expression [73,74]. These nucleic acids must be appropriately formulated to facilitate entry into target tissues, whether through physical, chemical, or biological methods. Among these, gene delivery mediated by engineered virions has proven to be one of the most successful approaches, despite potential immunological risks and limitations in “cargo” capacity. Therefore, proposing and evaluating alternative virions to traditional ones is necessary to support this growing industry. Among them, the baculovirus AcMNPV is proposed as an alternative to adenoviral vectors, as it allows transient expression with a superior immunological profile, with the advantages of large-scale manufacturing given the availability of insect cell lines that grow in suspension, and with the availability of culture media free of fetal bovine serum. However, since AcMNPV does not naturally infect mammals, it is crucial to generate optimized versions that provide better results.

In this work, we evaluated the addition of the vesicular stomatitis virus glycoprotein (VSV-G) to the AcMNPV BVs, studying how its differential presence, when expressed at various times during the infective cycle in insects, affects the infective and productive properties of virions and their effects on mammalian cells and rat hind limbs. We found that the expression of VSV-G from the immediate early to the final stages of the infective cycle does not affect the infective rate or productivity in the generation of viral progeny in insect cells growing in a monolayer, which could be an important limitation. Additionally, in all the tested cell lines, the pseudotyped BVs exhibited greater transducing capacity in mammalian cells, with maximum efficiency when VSV-G is expressed in the very late phase of the cycle. This virus also performed better than the non-pseudotyped one in the muscles of the rats to which it was administered, giving excellent levels of the transgene expression with only 10^4^ total infectious viral particles. These results, along with the other existing literature [50,51,52,53,54], demonstrate the usefulness of adding this glycoprotein to AcMNPV BVs for biotechnological applications in mammals. The choice of baculoviral promoter for pseudotyping is indeed critical, as recent evidence demonstrates that *polyhedrin*-driven VSV-G overexpression can compromise BV particle integrity and induce mammalian cell toxicity [75]. While our study did not explicitly assess BV morphology or cytotoxicity, we observed no adverse effects in transduced mammalian cells or rat muscle tissue—possibly due to the more moderate expression kinetics of the very late selected promoter compared to *polyhedrin*. Nevertheless, further studies should systematically evaluate these safety and structural parameters to confirm the optimal promoter selection for clinical applications.

As another future precaution, it will be important to analyze whether the generation of syncytia in the producing insect cells compromises BV production in cell suspension, as that approach is preferred in the industry. In such a case, choosing its milder and earlier expression (as with the *ie1* and *gp64* promoters) could be the best cost–benefit choice. If this is the choice for AcMNPV pseudotyping due to manufacturing advantages, their transient activity in mammalian cells must be carefully evaluated as a potential safety concern [76]. While this promoter-driven expression of VSV-G could theoretically enhance immunogenicity, similar risks are routinely managed in adenoviral vector therapies. As a precaution, insect packaging cell lines could be implemented to eliminate mammalian transcription entirely while preserving production benefits [53]. Undoubtedly, with future modifications to the AcMNPV genome, its relevance to the human pharmaceutical industry will increase.

## Figures and Tables

**Figure 1 vaccines-13-00693-f001:**
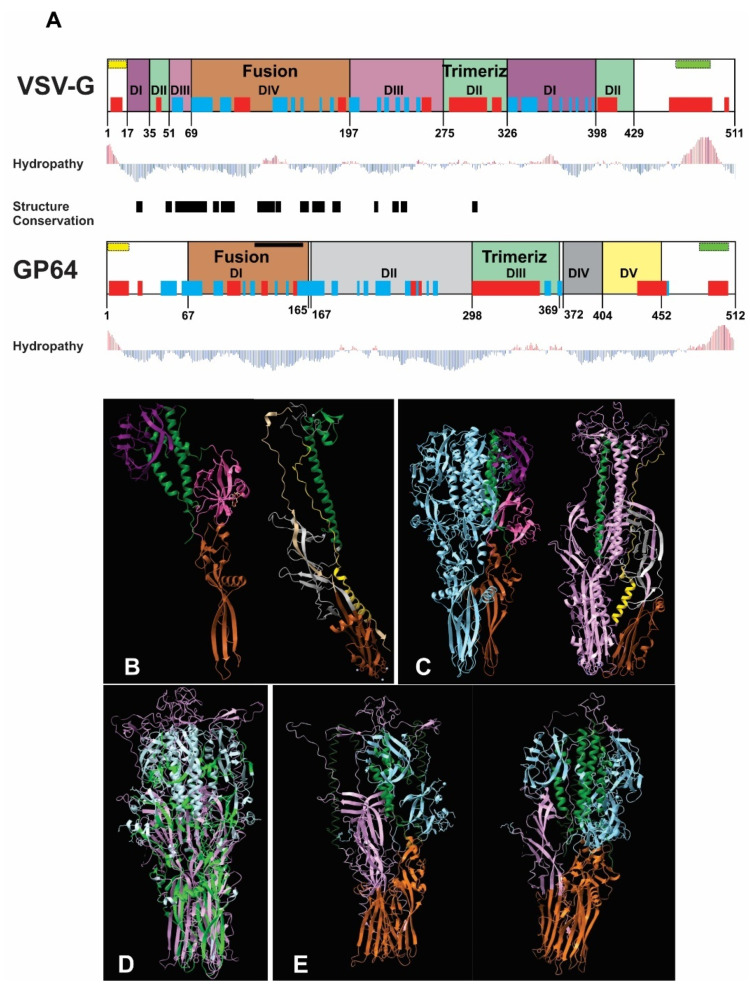
Bioinformatics comparison of VSV-G and GP64 structures and predicted heterotrimer formation: (**A**) Structural alignment of VSV-G (domains I–IV) and GP64 (domains I–V), with color-coded functional domains (fusion and trimerization domains share identical colors between proteins). Secondary structure elements (α-helices in red; β-sheets in blue), signal peptide (yellow), and transmembrane region (green) are shown above each domain. Hydropathy profiles (red bars: hydrophobic; blue bars: hydrophilic) and structurally conserved regions (black box) are indicated. (**B**) Monomeric structures with color-coded domains (fusion: brown; trimerization: green). (**C**) Homotrimeric structures (VSV-G: light blue; GP64: purple). (**D**) Structural superposition of homotrimers (VSV-G: light blue; GP64: pink) showing conserved regions (green) identified by the Dali server. (**E**) Predicted heterotrimer models (1GP64:2VSV-G and 2GP64:1VSV-G), with fusion and trimerization domains colored consistently with panels (**B**–**D**).

**Figure 2 vaccines-13-00693-f002:**
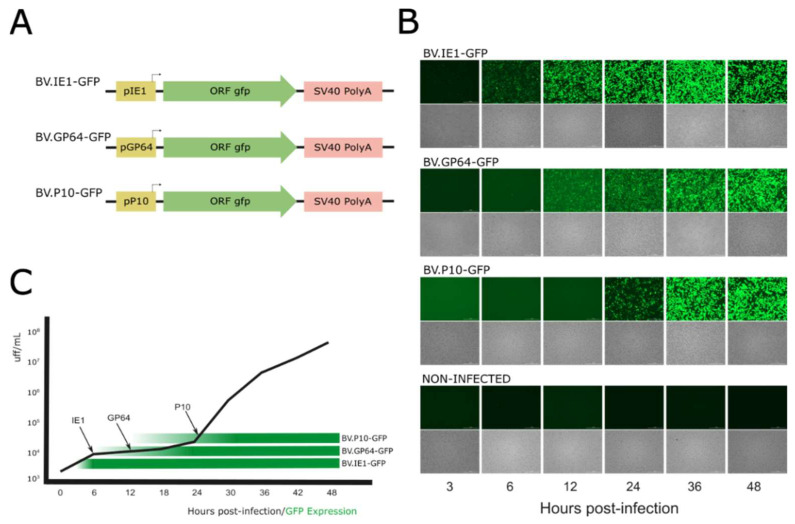
Time expression of AcMNPV *ie1*, *gp64*, and *p10* promoters: (**A**) Recombinant genes expressing GFP contained in three different AcMNPV. (**B**) Bright-field and GFP-filtered fluorescence microscopy images (25×) of the non-infected and infected Sf9 cells at different hours post-infection (hpi). (**C**) BV production kinetics overlapped with the GFP production driven by the different promoters. The black arrows indicate the times at which GFP was significantly detected by fluorescence microscopy, and the black line represents BV progeny in the conditioned medium (Foci-Forming Unit per mL; FFU/mL). The green horizontal bars mark the initial detection of GFP-expressing cells and serve as a visual reference only. These experiments were designed to determine the transcriptional onset timing at selected post-infection intervals (3–48 hpi), not to quantify comparative promoter strength or GFP expression levels.

**Figure 3 vaccines-13-00693-f003:**
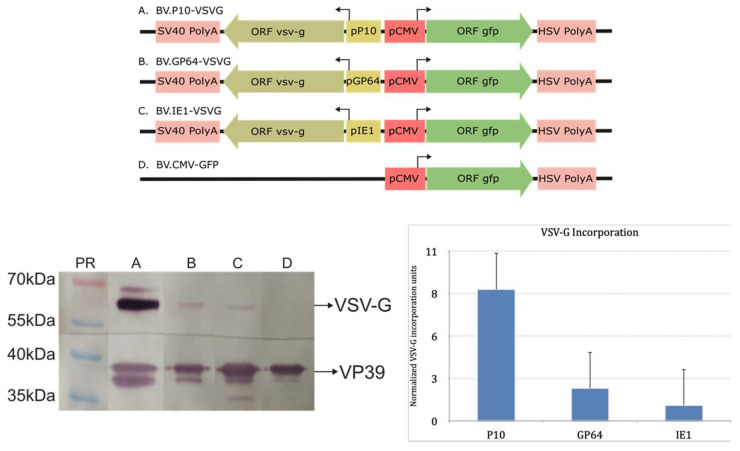
Immunodetection of VSV-G in recombinant AcMNPV: (**Top**) Recombinant genes expressing VSV-G in insects and GFP in mammals contained in four different baculoviruses (A: BV.P10-VSVG; B: BV.GP64-VSVG; C: BV.IE1-VSVG; D: BV.CMV-GFP). (**Bottom left**) Representative Western blot assay where ultraconcentrated samples of BVs were analyzed with anti-VSV-G and anti-VP39 antibodies. At the top of each lane, the identity of the sample studied is indicated. The most important protein sizes of molecular weight bands are indicated. (**Bottom right**) VSV-G incorporation plot obtained by the band densitometry of the Western blot assays using the ImageJ program. The VSV-G incorporation was normalized using the viral protein VP39 and then compared against the value obtained for BV.IE1-VSVG. Means and standard deviations are shown (n = 3). In the histogram, recombinant viruses are indicated by the promoters that govern VSV-G expression: P10 (BV.P10-VSVG); GP64 (BV.GP64-VSVG); and IE1 (BV.IE1-VSVG).

**Figure 4 vaccines-13-00693-f004:**
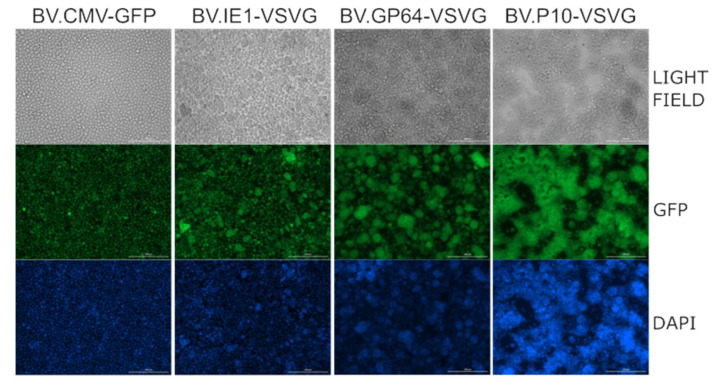
VSV-G activity as a syncytium former in insect cells. The recombinant BV.P10-VSVG, BV.GP64-VSVG, BV.IE1-VSVG, and BV.CMV-GFP were used to infect the Sf9 cells at an MOI of 5, which were then incubated in growth medium. Seventy-two hours post-infection, the cells were fixed and stained with 4′,6-diamidino-2-phenylindole (DAPI). Syncytium formation was analyzed by fluorescence microscopy (100×). Representative microscopy images are shown, where fusions between the infected cells can be observed.

**Figure 5 vaccines-13-00693-f005:**
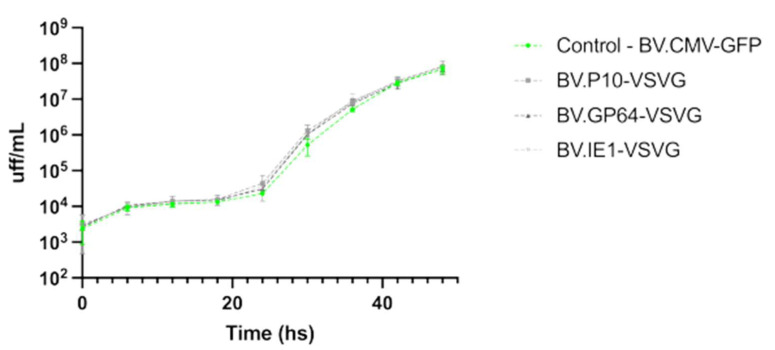
Performance of pseudotyped-BV yields throughout an infection cycle in insect cells. The recombinant BV.P10-VSVG, BV.GP64-VSVG, BV.IE1-VSVG, and BV.CMV-GFP were used to infect the Sf9 cells at an MOI of 5 and incubated with growth medium. Every 6 h (and for 48 h), the conditioned medium was recovered (and the cells were immediately washed to incorporate the same volume of fresh medium), from which the BV titer was subsequently determined as Foci-Forming Unit per mL (FFU/mL). The average values of 3 replicates and the corresponding standard deviation are indicated. Non-significant differences were detected (*p* < 0.05).

**Figure 6 vaccines-13-00693-f006:**
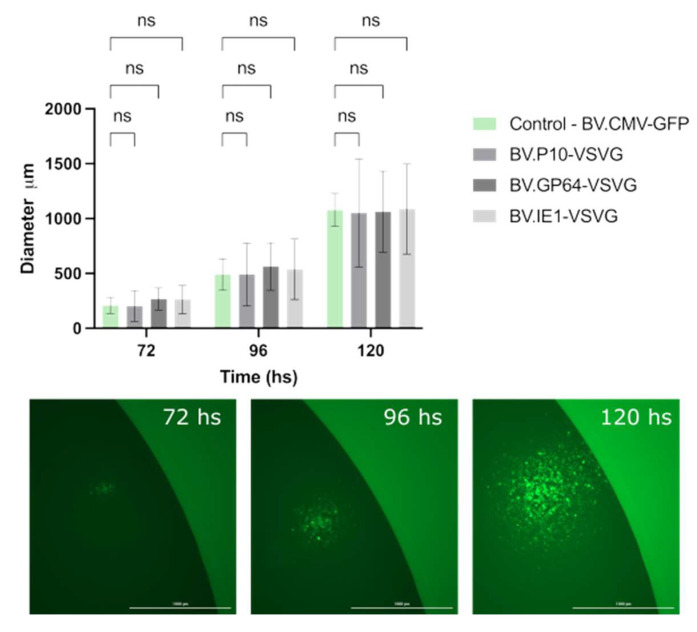
Performance of pseudotyped-BV infectivity throughout an infection cycle in insect cells. The recombinant BV.P10-VSVG, BV.GP64-VSVG, BV.IE1-VSVG, and BV.CMV-GFP were used to infect the Sf9 cells at an MOI of 5, which were then trapped in low-melting agarose matrix supplemented with growth medium. Every 24 h (and for 120 h), the infected monolayers were analyzed by fluorescence microscopy, and the diameter of infection foci was measured. (**Top**) The average values of 3 replicates and the corresponding standard deviation are indicated. Non-significant differences (ns) were detected (*p* < 0.05). (**Bottom**) Representative images (25×) of infection foci over time.

**Figure 7 vaccines-13-00693-f007:**
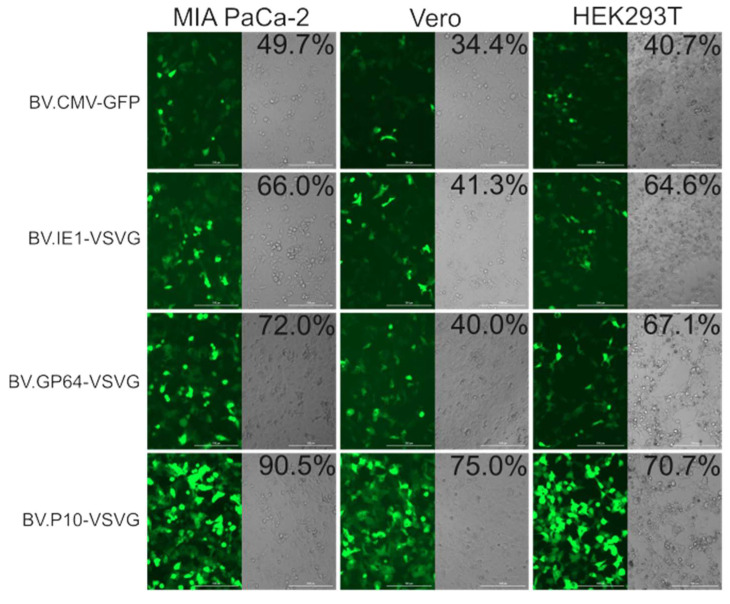
In vitro mammalian transduction assay. The cell lines MIA PaCa-2, Vero, and HEK293T were transduced with pseudotyped (BV.IE1-VSVG, BV.GP64-VSVG, and BV.P10-VSVG) and control (BV.CMV.GFP) baculoviruses at an MOI of 100, and after 24 h, analyzed by fluorescence microscopy (100×) and flow cytometer. The numbers indicated correspond to the GFP-expressing cell fraction determined by flow cytometry (performed in one of the replicates). Representative microscopy images of the results found in three replicates are shown using both bright-field microscopy (cell morphology) and fluorescence microscopy (GFP detection).

**Figure 8 vaccines-13-00693-f008:**
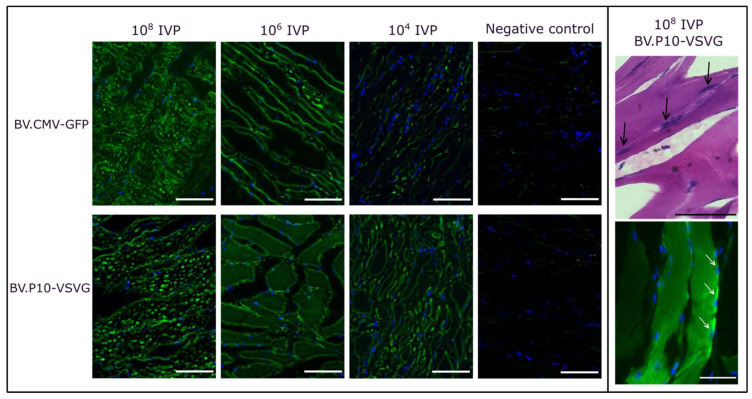
In vivo mammalian transduction assay. Rats were injected in the hind limbs with BV.CMV-GFP (control) or BV.P10-VSVG at three doses (10^8^, 10^6^, or 10^4^ total infectious viral particles [IVPs]). (**Left panel**) Tissues were harvested 3 days post-injection, and cryosections were stained with DAPI (nuclei) and analyzed by fluorescence microscopy (20× magnification). (**Right panel**) Cryosections of tissues from the rats injected with 10^8^ IVP of BV.P10-VSVG and stained with hematoxylin and eosin, or with DAPI (nuclei), analyzed by optical and fluorescence microscopy (40× magnification). The black (top photograh, Right panel) and white (bottom photograph, Right panel) arrows point to several nuclei showing the multinuclear nature of the skeletal muscle.

**Table 1 vaccines-13-00693-t001:** Primers used in PCR assays.

Primer Name	Sequence (5′ to 3′)
Fw.pIE1	CTCGAGTTGCACAACACTATTAT
Rv.pIE1	GGATCCAGTCACTTGGTTGTTCAC
Fw.pGP64	GGATCCCTTGCTTGTGTGTTCCTTATTG
Rv.pGP64	CTCGAGGATGACCACCTCCAG
FwVSVG	GGATCCGACACTATGAAGTGCC
RvVSVG	GATATCTGATTTGAGTTACTTTCC

**Table 2 vaccines-13-00693-t002:** Recombinant plasmid and viruses generated in this work.

**Plasmid Name**	**Gene Content**
pFastBacPromIE1/GFP	pFastBac-Dual backbone, prom-ie1/gfp/spA_SV40
pFastBacPromGP64/GFP	pFastBac-Dual backbone, prom-gp64/gfp/spA_SV40
pFastBacPromp10/GFP	pFastBac-Dual backbone, prom-p10/gfp/spA_SV40
pFastBacPromIE1/VSVG	pFastBac-Dual backbone, prom-ie1/VSV-G/spA_SV40 + prom-CMV/gfp/spA-HSV
pFastBacPromGP64/VSVG	pFastBac-Dual backbone, prom-gp64/VSV-G/spA_SV40 + prom-CMV/gfp/spA-HSV
pFastBacPromP10/VSVG	pFastBac-Dual backbone, prom-p10/VSV-G/spA-SV40 + prom-CMV/gfp/spA-HSV
**Baculovirus Name**	**Gene Content**
BV.IE1-GFP	bMON14272 backbone, prom-ie1/gfp/spA_SV40
BV.GP64-GFP	bMON14272 backbone, prom-gp64/gfp/spA_SV40
BV.P10-GFP	bMON14272 backbone, prom-p10/gfp/spA_SV40
BV.IE1-VSVG	bMON14272 backbone, prom-ie1/VSV-G/spA_SV40 + prom-CMV/gfp/spA-HSV
BV.GP64-VSVG	bMON14272 backbone, prom-gp64/VSV-G/spA_SV40 + prom-CMV/gfp/spA-HSV
BV.P10-VSVG	bMON14272 backbone, prom-p10/VSV-G/spA_SV40 + prom-CMV/gfp/spA-HSV
BV.CMV-GFP	bMON14272 backbone, prom-CMV/gfp/spA-HSV

Prom = promoter; spA = PolyA signal; SV40 = Simian Virus 40; HSV = Herpes Simplex Virus.

## Data Availability

Data are provided in the manuscript, and additional details will be provided upon request from the authors.

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
