# Peer review of "Early to Late VSV-G Expression in AcMNPV BV Enhances Transduction in Mammalian Cells but Does Not Affect Virion Yield in Insect Cells"

_vaccines, 2025, doi:10.3390/vaccines13070693_

Round 1
Reviewer 1 Report
Comments and Suggestions for Authors
The author examined whether expressing VSVG could enhance the efficiency of AcMNPV as MamBac. However, the following questions need to be answered:
- All recombinant AcMNPV were prepared using the Bac-to-Bac. Therefore, the GP64 gene in the AcMNPV genome exists. In this study, the three promoters used to drive expression of the VSV-G protein actually coexist with the GP64 protein. So, does this study consider that the VSV-G protein and the GP64 protein were packaged together with AcMNPV?
- If the author believes that the VSV-G protein is involved in the packaging process of AcMNPV, then the results of the Western Blot should not only show the content of VSV-G, but also the content of GP64. Moreover, it is expected that the contents of VSV-G and GP64 in the same viral quantity will be in a total balance state.
- In the Western blot experiment, the authors did not provide a detailed description of the experimental samples and how they were obtained in the methods section. If it was cell lysate, it would not be able to reflect the content of VSV-G protein in the virus. Because the cell lysate contains many free proteins. If it was purified virus, it should explain how it was purified and quantified.
- In Figure 2B, it can be seen from the green background of each picture that the author did not use the same exposure time or contrast when taking the photos. This is very unfavorable for the comparison of fluorescence intensity.The author needs to adjust the exposure time and contrast to be consistent, and provide the WB results of GFP at the corresponding time point as shown in Figure 1C to support the conclusion.
Author Response
We sincerely appreciate the reviewer's time and thoughtful evaluation of our work. Their constructive comments have significantly strengthened our manuscript. Responses to reviewer comments are provided in blue text beneath each observation.
Comments and Suggestions for Authors:
The author examined whether expressing VSVG could enhance the efficiency of AcMNPV as MamBac. However, the following questions need to be answered:
- All recombinant AcMNPV were prepared using the Bac-to-Bac. Therefore, the GP64 gene in the AcMNPV genome exists. In this study, the three promoters used to drive expression of the VSV-G protein actually coexist with the GP64 protein. So, does this study consider that the VSV-G protein and the GP64 protein were packaged together with AcMNPV?
As the reviewer rightly notes, GP64 is retained in the AcMNPV genome in the Bac-to-Bac system, resulting in co-expression with VSV-G. Bioinformatics analyses suggest that GP64 and VSV-G share structural similarities and could potentially form heterocomplexes when co-present in the viral envelope. Given this, we hypothesize that the relative abundance of VSV-G in budded virions (BVs) may vary depending on the promoter-driven expression kinetics (e.g., strength and timing). This differential incorporation could influence viral infectivity in insect cells and transduction efficiency in mammalian systems.
- If the author believes that the VSV-G protein is involved in the packaging process of AcMNPV, then the results of the Western Blot should not only show the content of VSV-G, but also the content of GP64. Moreover, it is expected that the contents of VSV-G and GP64 in the same viral quantity will be in a total balance state.
We agree with the reviewer that it would be interesting to analyze the amount of GP64 in the generated viral variants. However, unfortunately, we do not have these antibodies. Therefore, we decided to focus solely on the presence of VSV-G, normalizing with Vp39.
- In the Western blot experiment, the authors did not provide a detailed description of the experimental samples and how they were obtained in the methods section. If it was cell lysate, it would not be able to reflect the content of VSV-G protein in the virus. Because the cell lysate contains many free proteins. If it was purified virus, it should explain how it was purified and quantified.
We sincerely apologize for not specifying the number of virions analyzed in the Western blot. This detail has now been added to the revised Materials and Methods section. Regarding the reviewer’s concern about sample purity, we confirm that the Western blot was performed with ultra-concentrated BV particles, not cell lysates. As explicitly described in Section 2.4 of our original submission, our purification protocol (low-MOI production, 4-day harvest, clarification by centrifugation, and sucrose-cushion ultracentrifugation) is designed to prevent contamination with cellular debris or free VSV-G. This method has been rigorously validated in prior in vivo studies (DOIs: 10.1016/j.jcyt.2020.06.010, 10.1161/JAHA.123.031515), ensuring reliable virion preparations. We appreciate the opportunity to clarify this point and have revised the manuscript accordingly.
- In Figure 2B, it can be seen from the green background of each picture that the author did not use the same exposure time or contrast when taking the photos. This is very unfavorable for the comparison of fluorescence intensity.The author needs to adjust the exposure time and contrast to be consistent, and provide the WB results of GFP at the corresponding time point as shown in Figure 1C to support the conclusion.
We sincerely appreciate the reviewer’s careful observation regarding Figure 2B. We would like to clarify that the purpose of these fluorescence images was not to quantitatively compare GFP expression levels between time points, but rather to qualitatively demonstrate the first appearance of GFP-positive cells (regardless of fluorescence intensity) at each sampled time point (3, 6, 12, 24, 36, and 48 hours). This approach allowed us to determine the onset timing of promoter activity under our experimental conditions. To avoid any potential misunderstanding, we have added explicit clarification about the qualitative nature of these observations in both the manuscript text and figure caption. While we agree that quantitative Western blot data could provide complementary information, we respectfully note that such analysis would represent a different experimental approach than what was intended for this particular figure. Our goal here was specifically to track the temporal initiation of expression rather than measure expression levels. We hope these clarifications adequately address the reviewer’s concerns while maintaining the original purpose of this experimental approach.

Reviewer 2 Report
Comments and Suggestions for Authors
In the manuscript “Early to Late VSV-G expression in AcMPNV BV enhances transduction in mammalian cells but not affect virion yield in insects” by Simonin et al., the authors perform a promoter study test to evaluate baculovirus pseudotyping level and performance with VSV glycoprotein.
The manuscript identifies the need to investigate to optimise VSV-G expression kinetics in a putative GP64/VSVG heterotrimer formation which, in my opinion, is loosely grounded in reality and supported only by an in silico prediction presented in this work. Although the premise is wrong, the implementation is supported by data, although it will benefit from further experiments on some of the point raised.
From the title, it appears the manuscript is focusing on expression dynamics, but these cannot be separately assessed from expression levels especially in the baculovirus context. Hyperactive promoters as polH and p10 are inevitably producing viral progenies displaying higher amounts of glycoprotein, masking the effect of expression dynamics when compared with standard (non hyperactive) promoters as IE1 and gp64.
The choice of promoter has only been driven by expression dynamics, while expression levels have not been considered. The information was however available in a recent report which has not been cited (Bruder and Aucoin 2022) which details expression kinetics and levels for a range of BV promoters.
The authors also do not mention a recent report (Mattioli, Raele et al. 2024) which focused on analysing the effect of VSV-G overexpression levels on BV production and transduction properties on mammalian cells. In particular, polH driven expression caused structural defects on BV nanoparticles and excess toxicity in mammalian cells at high MOI. P10 is likely to produce similar effects and the authors should test for these phenotypes or, at the very least, discuss these aspects since toxicity and stability have not been tested in the present study.
The choice of promoter tested is additionally odd for another reason. As stated in text, early promoters do not require the viral RNA polymerase and exploit host factors for transcription. The authors are probably aware that early genes are expressed in mammalian cells, although transiently (Shin, Choi et al. 2020). Usage of IE1 will inevitably lead to VSV-G expression in mammalian cells, although transient. Gp64, although classified as a late gene, is atypical in having an early and late expression pattern and, as IE1, would drive expression in mammalian cells. In this context, the only safe viral promoter to use for mammalian cells transduction was p10 from the start.
Comments
Title – Please amend to “Early to Late VSV-G expression in AcMPNV BV enhances transduction in mammalian cells but do not affect virion yield in insect cells”.
Line 19 – Optimal VSV-G expression strategies have been already investigated in a recent study (Mattioli, Raele et al. 2024), not cited in text. This reduces the novelty around the current manuscript and the authors should put emphasis on expression timings instead, although arguably these cannot be effectively decoupled from expression levels.
Line 41 – “Which are infective per se”, please add “in insects”.
Line 46 – Minor comment - “In the baculoviral species that generate them” is dispensable. Although some baculovirus might have a life cycle that does not involve ODV formation, this does not apply to AcMNPV and it could be confusing if mentioned here.
Line 51 – Minor comment - The nomenclature Alphabaculovirus aucalifornicae although correct, is not widely adopted. This term can be omitted.
Line 58 – “Injection of the viral genome”. The accepted viral entry stages are: attachment, entry, endosomal escape, nuclear entry and uncoating. Genome injection is not a recognised step in baculovirus life cycle, this is more reminiscent of phage biology and it would be inappropriate to include it here.
Line 87 – “Don’t” please change to “do not”.
Line 95 – The authors note that transient gene expression is a limitation. While this is true for classical gene therapy approach, in the era of genome editing this is rather advantageous when the whole delivery field is moving towards delivery vehicles with short persistence leaving behind permanent genome changes. In this context the transient expression is a desirable feature. I leave to the authors consideration on whether or not expand on this point.
Line 98 – With some important distinction where a gp64 target might be absent, entry is generally very efficient. The major problem is the inability of gp64 to escape the endosomes, as pointed by one of the references cited in this work. In this context VSV-G is particularly helpful since it engages in post-fusion structures at a pH which is physiologically found in early endosomes. Indeed, gp64 pseudotyped viruses can be enhanced by using small-molecule endosome acidifiers as ouabain, demonstrating that entry is not a problem, but endosomal escape is.
Comment – VSV-G and gp64 comparative studies
The rationale behind this paragraph is not entirely clear to me. Comparisons between gp64 and VSV-G structures have been done elsewhere, but the authors suggest that, based on structure similarities, there is a chance that gp64 and VSV-G could form heterotrimers. As a consequence of a putative heterotrimerisation, the authors further speculate that this could compromise virion productivity in insect cells. While the hypothesis is suggestive, I found that it does not take into account previous literature. Numerous reports have demonstrated that VSV-G pseudotyping using polH promoter (which is arguably the strongest viral promoter) do not affect virus production in insect cells. In my opinion, this automatically abrogates the need to investigate potential issues linked to putative heterotrimers formation. Additionally, the authors provide a bioinformatic analysis in support of their hypothesis but there is no wet-lab data supporting the validity of the prediction. The whole paragraph and figure, remains highly speculative and it does not add anything to the manuscript.
Unless proven with wet-lab experiments, the existence of VSV-G/GP64 trimers remains highly speculative, but also unlikely due to the underpinning biology of GP64 trimer formation which requires intramolecular disulfide bonds, contrary to VSV-G. Additionally, when not assembled into trimers, GP64 monomers half-life is extremely short, and unlikely to be fostering trimerization with another glycoprotein.
Line 329 – The authors should consider whether to put emphasis on expression levels or expression dynamics. As discussed, expression levels have been previously investigated, and throughout the text, there seems to be more emphasis on expression kinetics, with a large section dedicated to the description of early, late and very late stages of gene expression.
Line 379 – A traditional syncytium formation requires a short pH acidification. Rather than a syncytium assay, this is an observation of spontaneous syncytium formation following VSV-G expression. This happens because the pH (6.4) of in insect cell medium, is already sufficient to trigger VSV-G post-fusion transition.
Figure 3 – How many times was the wester blot repeated? Please provide this information in the figure legend. The amount of virus on the blot seem to be uneven, based on vp39 intensity per lane. It would be helpful to provide a panel with estimated VSV-G incorporation levels normalised for vp39 intensities. Although it is clear that p10 provides the highest incorporation rate, gp64 seems to produce much better incorporation rates than VSV-G. The data are similar to a previous report (Mattioli, Raele et al. 2024) which showed that polH was massively outperforming other promoters in VSV-G incorporation.
Line 384 – in addition to this reference, it would helpful to cite papers which have previously described VSVG pseudotyping in the context of gp64 baculoviruses. These works have repeatedly demonstrated that titres in insect cells were not compromised.
Figure 5 – Significant differences, reflecting expression dynamics, would be probably found if titration was performed using end-point dilution on mammalian cells.
Figure 7 – How many times was this experiment repeated? There is indication of flow-cytometry data in the figure legend but there is no standard deviation of the values represented. Please include a histogram plot with error bars for flow-cytometry data, including a plot of MFI if relevant.
The experiment has additionally been performed only for one fixed (and very high) MOI. The experiment should be performed with serial dilutions to estimate TU/mL which would more accurately define the transduction power of each pseudotyping cassette. Remarkably, in HEK293T and MIA PaCa2, the transduction levels are enhanced to a more similar extent. This is in line with previous findings describing that VSV-G overexpression is not absolutely required to optimise transduction.
The bright-field pictures are undiscernible, but it looks like HEK293T are in a particular bad state when p10 promoter is used. Have the authors checked toxicity stemming from VSV-G hyper-display? This could be derived from gating of flow-cytometry data or it should be assessed if not investigated.
References
Bruder, M. R. and M. G. Aucoin (2022) "Utility of Alternative Promoters for Foreign Gene Expression Using the Baculovirus Expression Vector System." Viruses 14 DOI: 10.3390/v14122670.
Mattioli, M., R. A. Raele, G. Gautam, U. Borucu, C. Schaffitzel, F. Aulicino and I. Berger (2024) "Tuning VSV-G Expression Improves Baculovirus Integrity, Stability and Mammalian Cell Transduction Efficiency." Viruses 16 DOI: 10.3390/v16091475.
Shin, H. Y., H. Choi, N. Kim, N. Park, H. Kim, J. Kim and Y. B. Kim (2020) "Unraveling the Genome-Wide Impact of Recombinant Baculovirus Infection in Mammalian Cells for Gene Delivery." Genes 11 DOI: 10.3390/genes11111306.
Author Response
We sincerely appreciate the reviewer's time and thoughtful evaluation of our work. Their constructive comments have significantly strengthened our manuscript. Responses to reviewer comments are provided in blue text beneath each observation.
Comments and Suggestions for Authors:
In the manuscript “Early to Late VSV-G expression in AcMPNV BV enhances transduction in mammalian cells but not affect virion yield in insects” by Simonin et al., the authors perform a promoter study test to evaluate baculovirus pseudotyping level and performance with VSV glycoprotein.
The manuscript identifies the need to investigate to optimise VSV-G expression kinetics in a putative GP64/VSVG heterotrimer formation which, in my opinion, is loosely grounded in reality and supported only by an in silico prediction presented in this work. Although the premise is wrong, the implementation is supported by data, although it will benefit from further experiments on some of the point raised.
We appreciate the reviewer's thoughtful critique of our hypothesis regarding potential GP64/VSV-G heterotrimer formation. While we acknowledge that our in silico modeling represents an initial theoretical framework, we would like to clarify that our experimental design was specifically focused on investigating how differential VSV-G expression timing affects infection and transduction efficiency in insect and mammalian cells, respectively.
The possible competition between GP64 and VSV-G for membrane localization suggests meaningful biological interactions warranting further study. We agree that additional biochemical characterization would strengthen the heterotrimer hypothesis, and we are currently initiating specific studies to directly test this. However, as these investigations would constitute a separate, substantial body of work, we believe the current manuscript appropriately focuses on the demonstrable phenotypic consequences of regulated VSV-G expression.
From the title, it appears the manuscript is focusing on expression dynamics, but these cannot be separately assessed from expression levels especially in the baculovirus context. Hyperactive promoters as polH and p10 are inevitably producing viral progenies displaying higher amounts of glycoprotein, masking the effect of expression dynamics when compared with standard (non hyperactive) promoters as IE1 and gp64.
We appreciate the reviewer's insightful comment regarding the interplay between expression dynamics and levels in baculovirus systems. While we agree that promoter strength significantly impacts glycoprotein incorporation, our study specifically investigates how temporal expression patterns influence VSV-G functionality, independent of absolute expression levels.
Some consideration supports our approach. Ie1 promoter-driven VSV-G benefits from intact cellular machinery (e.g., ER/Golgi function) during initial infection stages, potentially yielding better glycosylated, more functional glycoproteins despite lower total expression compared to very late promoters. During late infection (when p10/polh are active), cellular resources are prioritized for ODV/OB production, which may compromise BV envelope protein quality even at high expression levels. We agree that promoter strength is an important variable, but maintain that expression timing represents an independent parameter worthy of investigation. We introduced these clarifications into the manuscript.
The choice of promoter has only been driven by expression dynamics, while expression levels have not been considered. The information was however available in a recent report which has not been cited (Bruder and Aucoin 2022) which details expression kinetics and levels for a range of BV promoters.
We thank the reviewer for bringing this important reference to our attention. We have now incorporated the citation to Bruder and Aucoin (2022) in our discussion of promoter selection criteria, along with additional commentary about how their findings on both expression levels and kinetics inform our experimental design. This valuable addition strengthens the context for our promoter choice rationale.
The authors also do not mention a recent report (Mattioli, Raele et al. 2024) which focused on analysing the effect of VSV-G overexpression levels on BV production and transduction properties on mammalian cells. In particular, polH driven expression caused structural defects on BV nanoparticles and excess toxicity in mammalian cells at high MOI. P10 is likely to produce similar effects and the authors should test for these phenotypes or, at the very least, discuss these aspects since toxicity and stability have not been tested in the present study.
We thank the reviewer for highlighting the important findings of Mattioli et al. (2024) regarding polyhedrin-driven VSV-G toxicity and BV instability. While our study focused on comparing transduction efficiencies across promoters (rather than structural or cytotoxic effects), we acknowledge the need to address these safety concerns in future work. We have now incorporated a discussion of these risks in the manuscript, emphasizing that our selected promoters (ie1, gp64, p10) may mitigate such issues due to their distinct expression dynamics. Should p10-driven expression prove problematic in subsequent studies, alternative strategies—such as promoter engineering or insect packaging cell lines—could be explored to balance yield and safety.
The choice of promoter tested is additionally odd for another reason. As stated in text, early promoters do not require the viral RNA polymerase and exploit host factors for transcription. The authors are probably aware that early genes are expressed in mammalian cells, although transiently (Shin, Choi et al. 2020). Usage of IE1 will inevitably lead to VSV-G expression in mammalian cells, although transient. Gp64, although classified as a late gene, is atypical in having an early and late expression pattern and, as IE1, would drive expression in mammalian cells. In this context, the only safe viral promoter to use for mammalian cells transduction was p10 from the start.
We appreciate the reviewer’s insightful comment about promoter activity in mammalian cells. While we acknowledge that early promoters (ie1/gp64) can drive transient expression in mammalian systems (as noted by Shin et al., 2020), their use in our study serves two purposes: (1) to compare BV pseudotyping efficiency across promoters with distinct dynamics, and (2) to evaluate how differential VSV-G quantities/qualities impact transduction. Importantly, any mammalian-cell expression of baculoviral proteins (including VSV-G) would likely mirror the behavior of clinical adenoviral vectors, which also express viral genes alongside transgenes—a well-managed safety consideration in gene therapy. Should mammalian expression prove problematic, packaging cell lines could eliminate this concern while retaining pseudotyping benefits. We have expanded the Discussion to address these points, including the safety advantages of p10 as a late promoter, and have added the suggested citation to contextualize promoter cross-activity.
Comments
Title – Please amend to “Early to Late VSV-G expression in AcMPNV BV enhances transduction in mammalian cells but do not affect virion yield in insect cells”.
Thank you for the correction. The title has been modified.
Line 19 – Optimal VSV-G expression strategies have been already investigated in a recent study (Mattioli, Raele et al. 2024), not cited in text. This reduces the novelty around the current manuscript and the authors should put emphasis on expression timings instead, although arguably these cannot be effectively decoupled from expression levels.
We appreciate this observation and have now incorporated the suggested reference (Mattioli et al., 2024) into our discussion. While their work valuably examines VSV-G expression optimization, our study specifically focuses on the temporal dimension of expression - comparing early versus late promoter strategies - which provides distinct mechanistic insights into baculovirus pseudotyping.
Line 41 – “Which are infective per se”, please add “in insects”.
Thank you for the comment. To be more concise, given the great diversity of baculovirus species and, in many cases, their narrow host range, we introduced "in susceptible hosts".
Line 46 – Minor comment - “In the baculoviral species that generate them” is dispensable. Although some baculovirus might have a life cycle that does not involve ODV formation, this does not apply to AcMNPV and it could be confusing if mentioned here.
Thank you for your thoughtful comment. We agree with the reviewer’s observation regarding the specificity of AcMNPV’s life cycle, which invariably involves ODV and BV formation. However, in the text, we are referring specifically to the BV morphotype, which is not produced by all baculoviruses (e.g., gammabaculoviruses). In our previous work, reviewers have encouraged us to highlight this diversity, even when discussing AcMNPV, as it underscores that BVs are not universally present across the Baculoviridae family. Additionally, BV composition varies among genera: while betabaculoviruses and deltabaculoviruses rely solely on the F protein for fusion, GP64 is exclusive to group I alphabaculoviruses like AcMNPV. Thus, we included the note to avoid generalizations about BV biology and to emphasize the relevance of BVs for mammalian cell transduction. We appreciate the opportunity to clarify this point.
Line 51 – Minor comment - The nomenclature Alphabaculovirus aucalifornicae although correct, is not widely adopted. This term can be omitted.
We appreciate the reviewer's comment regarding nomenclature usage. Consistent with previous feedback from ICTV experts and other reviewers, we strive to adhere to official taxonomic standards in our publications. While we acknowledge that "Alphabaculovirus aucalifornicae" may not yet be universally adopted in the literature, we have retained the term to align with current ICTV classification guidelines. We are happy to adjust this if the editor considers it preferable for broader readability.
Line 58 – “Injection of the viral genome”. The accepted viral entry stages are: attachment, entry, endosomal escape, nuclear entry and uncoating. Genome injection is not a recognised step in baculovirus life cycle, this is more reminiscent of phage biology and it would be inappropriate to include it here.
Thank you for the correction. We've incorporated it into the text.
Line 87 – “Don’t” please change to “do not”.
Thank you for the correction. We've incorporated it into the text.
Line 95 – The authors note that transient gene expression is a limitation. While this is true for classical gene therapy approach, in the era of genome editing this is rather advantageous when the whole delivery field is moving towards delivery vehicles with short persistence leaving behind permanent genome changes. In this context the transient expression is a desirable feature. I leave to the authors consideration on whether or not expand on this point.
Thank you for this valuable perspective. We fully agree that transient expression can be advantageous for genome editing applications, offering enhanced safety control. We have revised the text to better reflect this dual nature of transient expression as both a potential limitation and benefit, depending on the therapeutic context.
Line 98 – With some important distinction where a gp64 target might be absent, entry is generally very efficient. The major problem is the inability of gp64 to escape the endosomes, as pointed by one of the references cited in this work. In this context VSV-G is particularly helpful since it engages in post-fusion structures at a pH which is physiologically found in early endosomes. Indeed, gp64 pseudotyped viruses can be enhanced by using small-molecule endosome acidifiers as ouabain, demonstrating that entry is not a problem, but endosomal escape is.
We fully agree with the reviewer’s insightful observation regarding GP64’s limitations in endosomal escape—a point well-supported by prior literature, including our cited references. As the reviewer notes, VSV-G’s ability to fuse at early endosomal pH addresses this bottleneck, while GP64-mediated entry remains efficient. To clarify this focus, we have amended the introduction to explicitly state that limitations in cell entry include the endosomal release process.
Comment – VSV-G and gp64 comparative studies
The rationale behind this paragraph is not entirely clear to me. Comparisons between gp64 and VSV-G structures have been done elsewhere, but the authors suggest that, based on structure similarities, there is a chance that gp64 and VSV-G could form heterotrimers. As a consequence of a putative heterotrimerisation, the authors further speculate that this could compromise virion productivity in insect cells. While the hypothesis is suggestive, I found that it does not take into account previous literature. Numerous reports have demonstrated that VSV-G pseudotyping using polH promoter (which is arguably the strongest viral promoter) do not affect virus production in insect cells. In my opinion, this automatically abrogates the need to investigate potential issues linked to putative heterotrimers formation. Additionally, the authors provide a bioinformatic analysis in support of their hypothesis but there is no wet-lab data supporting the validity of the prediction. The whole paragraph and figure, remains highly speculative and it does not add anything to the manuscript.
Unless proven with wet-lab experiments, the existence of VSV-G/GP64 trimers remains highly speculative, but also unlikely due to the underpinning biology of GP64 trimer formation which requires intramolecular disulfide bonds, contrary to VSV-G. Additionally, when not assembled into trimers, GP64 monomers half-life is extremely short, and unlikely to be fostering trimerization with another glycoprotein.
We sincerely appreciate the reviewer's thorough critique, which highlights limitations in our current speculative approach. We agree that the formation of VSV-G/gp64 heterotrimers remains biologically undetermined given the distinct assembly mechanisms of both glycoproteins (e.g., gp64’s dependence on intramolecular disulfide bonds). However, our bioinformatic analysis—employing novel computational tools not previously applied to this specific comparison—reveals unexpected structural similarities that merit further exploration, especially given the differential temporal expression of VSV-G under early vs. late promoters. While prior studies using the polyhedrin promoter show no apparent impact on virion productivity, we posit that syncytia formation (driven by VSV-G’s fusogenic activity) could pose scalability challenges in industrial manufacturing contexts. This hypothesis, coupled with the potential for altered glycoprotein distribution in BV envelopes due to promoter-driven timing effects, motivated our preliminary structural analysis.
In direct response to the reviewer's concerns, we have modified the paragraph to more clearly frame this as a preliminary bioinformatic exploration rather than an established biological phenomenon and highlighted that wet-lab experiments (currently underway) are needed to validate any structural predictions.
While we agree that heterotrimer formation remains improbable, we believe documenting these structural observations in the context of temporal expression differences provides value to the field, particularly for researchers exploring similar pseudotyping approaches.
Line 329 – The authors should consider whether to put emphasis on expression levels or expression dynamics. As discussed, expression levels have been previously investigated, and throughout the text, there seems to be more emphasis on expression kinetics, with a large section dedicated to the description of early, late and very late stages of gene expression.
Thank you for your comment. As the reviewer notes, we aim to analyze how BVs carrying varying quantities (and potentially qualities) of VSV-G infect insect cells and transduce mammalian cells, using different baculoviral promoters with distinct expression dynamics. We have implemented clarifications throughout the manuscript to better reflect this focus.
Line 379 – A traditional syncytium formation requires a short pH acidification. Rather than a syncytium assay, this is an observation of spontaneous syncytium formation following VSV-G expression. This happens because the pH (6.4) of in insect cell medium, is already sufficient to trigger VSV-G post-fusion transition.
We agree with the reviewer's accurate observation about the medium pH (6.4) enabling spontaneous VSV-G-mediated syncytium formation in insect cell cultures. Indeed, this phenomenon serves as a qualitative indicator of differential VSV-G incorporation in membranes of cells infected by our recombinant viruses - the same membranes from which budded virions (BVs) emerge. Importantly, these observations correlate well with our Western blot data showing varying VSV-G expression levels, validating our experimental approach. From a biotechnological perspective, this pH-dependent fusion activity could significantly impact industrial-scale production of VSV-G-pseudotyped BVs (or other fusogenic glycoprotein combinations), particularly when using late promoters in traditional Sf9 culture media. As we discuss in our conclusions, the cost-benefit analysis may favor earlier promoters that provide better control over this fusion activity while maintaining sufficient pseudotyping efficiency.
Figure 3 – How many times was the wester blot repeated? Please provide this information in the figure legend. The amount of virus on the blot seem to be uneven, based on vp39 intensity per lane. It would be helpful to provide a panel with estimated VSV-G incorporation levels normalised for vp39 intensities. Although it is clear that p10 provides the highest incorporation rate, gp64 seems to produce much better incorporation rates than VSV-G. The data are similar to a previous report (Mattioli, Raele et al. 2024) which showed that polH was massively outperforming other promoters in VSV-G incorporation.
We appreciate these insightful comments. The revised figure now specifies replicate numbers and includes normalized densitometry data (VSV-G/VP39). While confirming known promoter efficiency trends (Mattioli et al., 2024), our work extends these findings by linking incorporation levels to functional outcomes in both insect and mammalian systems, as demonstrated in later assays. All viral loads were carefully normalized by titration before loading.
Line 384 – in addition to this reference, it would helpful to cite papers which have previously described VSVG pseudotyping in the context of gp64 baculoviruses. These works have repeatedly demonstrated that titres in insect cells were not compromised.
We appreciate the reviewer’s comment. As suggested, we have now clarified this point in the text by citing relevant studies on VSVG pseudotyping in gp64 baculoviruses, which consistently show that insect cell titers remain unaffected. These works further support our discussion, as they highlight the importance of preserving gp64 expression due to the technical challenges of its elimination and the negative impacts this may have on production efficiency. We have incorporated this clarification into the manuscript.
Figure 5 – Significant differences, reflecting expression dynamics, would be probably found if titration was performed using end-point dilution on mammalian cells.
We agree with the reviewer that endpoint dilution titration on mammalian cells could reveal additional differences in expression dynamics. However, our experimental design prioritized (1) evaluating BV production parameters in insect cells (the primary system for manufacturing pseudotyped virions) and (2) assessing functional transduction efficiency in mammalian systems rather than detailed kinetic comparisons. This approach aligns with our study's focus on translational applicability.
Figure 7 – How many times was this experiment repeated? There is indication of flow-cytometry data in the figure legend but there is no standard deviation of the values represented. Please include a histogram plot with error bars for flow-cytometry data, including a plot of MFI if relevant.
The mammalian cell transduction assays were performed in three independent replicates, with a representative image shown in the figure. Flow cytometry analysis was conducted on one replicate, as all replicates yielded qualitatively consistent results. For transparency, we have now included the complete flow cytometry dataset in the additional material, which contains details of all the experiments performed.
The experiment has additionally been performed only for one fixed (and very high) MOI. The experiment should be performed with serial dilutions to estimate TU/mL which would more accurately define the transduction power of each pseudotyping cassette. Remarkably, in HEK293T and MIA PaCa2, the transduction levels are enhanced to a more similar extent. This is in line with previous findings describing that VSV-G overexpression is not absolutely required to optimise transduction.
We appreciate the reviewer’s suggestion regarding MOI optimization. While serial dilutions would indeed provide precise TU/mL measurements, our use of a fixed high MOI (consistent with values from our prior mammalian gene therapy studies -DOIs: 10.1016/j.jcyt.2020.06.010, 10.1161/JAHA.123.031515-) was intentional: it reflects clinically relevant transduction conditions for therapeutic applications, where maximal payload delivery is often prioritized. This approach aligns with established literature on pseudotyped BV efficacy testing and mirrors real-world translational scenarios. However, we acknowledge that dose-response data could offer additional mechanistic insights, and we have now included this limitation in the manuscript.
The bright-field pictures are undiscernible, but it looks like HEK293T are in a particular bad state when p10 promoter is used. Have the authors checked toxicity stemming from VSV-G hyper-display? This could be derived from gating of flow-cytometry data or it should be assessed if not investigated.
We appreciate the reviewer’s attention to potential cytotoxicity. While our qualitative assessments (bright-field imaging and in vivo observations) revealed no overt signs of toxicity—even with p10-driven VSV-G expression—we acknowledge that dedicated safety assays (e.g., flow cytometry gating for apoptosis/necrosis, LDH release) would provide more definitive conclusions. We have now explicitly noted this limitation in the manuscript and highlighted the need for systematic toxicity profiling in future work, particularly for hyperactive promoters like p10.
References
Bruder, M. R. and M. G. Aucoin (2022) "Utility of Alternative Promoters for Foreign Gene Expression Using the Baculovirus Expression Vector System." Viruses 14 DOI: 10.3390/v14122670.
Mattioli, M., R. A. Raele, G. Gautam, U. Borucu, C. Schaffitzel, F. Aulicino and I. Berger (2024) "Tuning VSV-G Expression Improves Baculovirus Integrity, Stability and Mammalian Cell Transduction Efficiency." Viruses 16 DOI: 10.3390/v16091475.
Shin, H. Y., H. Choi, N. Kim, N. Park, H. Kim, J. Kim and Y. B. Kim (2020) "Unraveling the Genome-Wide Impact of Recombinant Baculovirus Infection in Mammalian Cells for Gene Delivery." Genes 11 DOI: 10.3390/genes11111306.
Thank you for the references. All were incorporated into the manuscript.

Reviewer 3 Report
Comments and Suggestions for Authors
This paper describes the pseudo-typing of baculovirus, expressing GFP under the CMV promoter, with VSV-G under control of either ie1, gp64 or p10 promoters to determine if early, late or very late promoters, respectively, are the best option.
The introduction is quite long but does provide a good background for those not familair with baculovirus. However, the information provided in the introduction about the timing of gene expression for the early, late and very late phases of gene expression do not align with the data obtained in the results - partly because the introduction is refering to promoter activity and the results are looking at the accumulation of GFP protein over time as a proxy for promoter activity. The authors should make this much clearer in the results section. For example, in the introduction it is stated that very later promoter activity is maximal at 18-24 hpi and in the results GFP under p10 is only just begging to be expressed.
Overall as the authors themselves state several times, the results they have observed are those that would be expected - using a strong promoter give more VSV pseudo typing. The interesting and useful information for those developing gene therapy vectors is that this does not compromise BV production nor target gene expression - at least with the reporter GFP. The transduction of rat tissue was very interesting but the lack of detail provided made it hard to understand fully what was being presented.
If the following are addressed, the paper will provide a useful addition to the literature on the use of baculoviruses for gene therapy.
In line 75, the authors indicate there are several baculovirus genes classified as 'very late', in addition to p10 and polyhedrin. Please could they give some examples with references or modify the statement to say the very late genes are p10 and polyhedrin.
Line 127 - should be GP64 not Gp64.
In order to use baculovirus as gene therapy vectors for human use, all work must be carried out in animal-free medium so I wonder why in their studies the authors use insect cells gorwn in medium requiring 10% serum? Do they think their results would translate to using baculovirus amplified in animal-free medium? Perhaps this should be mentioned in the conclusion?
A general comment is that some of the figure legends are very brief and sometimes I had to refer to Materials and Methods to under them.
In Figure 2, a description of Panel C is missing, please could this be added.
In Figure 3, the blot is clearly a hybrid cut and pasted from several different blots, since there is a comparison being made between VSV-G in different constructs how did the authors ensure that loadings on the gel etc were comparable between blots - the VP39 loading is not consistent between tracks shown. What is the doublet in tracks A -C?
Figure 7 line 442 replace 'exposed' with 'transfected'? Were these representative images of those observed?
Figure 8 line 448 Please explain more clearly what tissues are being viewed - a histological stained section may be useful to provide context of what is being look at? Are these muscle tissues?? Arrows to point to key features would also be useful. The transduction of rat tissue is one of the more interesting parts of this paper and this figure tends to understate the results. A little more detail in the results/materials and methods would be helpful too.
Author Response
We sincerely appreciate the reviewer's time and thoughtful evaluation of our work. Their constructive comments have significantly strengthened our manuscript. Responses to reviewer comments are provided in blue text beneath each observation.
Comments and Suggestions for Authors:
This paper describes the pseudo-typing of baculovirus, expressing GFP under the CMV promoter, with VSV-G under control of either ie1, gp64 or p10 promoters to determine if early, late or very late promoters, respectively, are the best option.
The introduction is quite long but does provide a good background for those not familair with baculovirus. However, the information provided in the introduction about the timing of gene expression for the early, late and very late phases of gene expression do not align with the data obtained in the results - partly because the introduction is refering to promoter activity and the results are looking at the accumulation of GFP protein over time as a proxy for promoter activity. The authors should make this much clearer in the results section. For example, in the introduction it is stated that very later promoter activity is maximal at 18-24 hpi and in the results GFP under p10 is only just begging to be expressed.
We appreciate this observation. We've clarified in the text that GFP accumulation (our readout) lags behind transcriptional activity due to post-transcriptional delays, and adjusted figure legend accordingly.
Overall as the authors themselves state several times, the results they have observed are those that would be expected - using a strong promoter give more VSV pseudo typing. The interesting and useful information for those developing gene therapy vectors is that this does not compromise BV production nor target gene expression - at least with the reporter GFP. The transduction of rat tissue was very interesting but the lack of detail provided made it hard to understand fully what was being presented.
We thank the reviewer for underscoring our main finding: strong promoters enhance pseudotyping without compromising production/expression (except the formation of syncises in insect cells). For the in vivo data, we’ve added H&E-stained sections and expanded methodological details (Fig. 8) to improve clarity. These changes better highlight our goal: optimizing VSV-G incorporation for mammalian transduction while preserving insect-cell manufacturing efficiency.
If the following are addressed, the paper will provide a useful addition to the literature on the use of baculoviruses for gene therapy.
We sincerely appreciate the reviewer's constructive feedback and their recognition of our study's potential contribution to the baculovirus gene therapy field. We have carefully addressed all raised concerns, including those from other reviewers, and believe these revisions have significantly strengthened the manuscript's rigor and clarity.
In line 75, the authors indicate there are several baculovirus genes classified as 'very late', in addition to p10 and polyhedrin. Please could they give some examples with references or modify the statement to say the very late genes are p10 and polyhedrin.
Thank you for the comment. The entire paragraph that mentions what the reviewer pointed out has been modified.
Line 127 - should be GP64 not Gp64.
Thank you for the correction. We've incorporated it into the text.
In order to use baculovirus as gene therapy vectors for human use, all work must be carried out in animal-free medium so I wonder why in their studies the authors use insect cells gorwn in medium requiring 10% serum? Do they think their results would translate to using baculovirus amplified in animal-free medium? Perhaps this should be mentioned in the conclusion?
We appreciate the reviewer's important observation regarding serum-free media requirements for clinical applications. While our current study utilized GRACE´s medium with FBS (virus-filtered to remove potential adventitious agents), we fully acknowledge that serum-free alternatives exist for the baculovirus-insect cell system - a key advantage of this platform. We agree this translational consideration merits discussion and have therefore addressed the implications of animal-free production for therapeutic applications in the conclusions section.
A general comment is that some of the figure legends are very brief and sometimes I had to refer to Materials and Methods to under them.
We thank the reviewer for this important observation. We have comprehensively revised all figure legends to provide complete methodological details, eliminating the need to consult the Materials and Methods section for interpretation. Specific experimental conditions and technical specifications have now been incorporated throughout.
In Figure 2, a description of Panel C is missing, please could this be added.
We apologize for omitting part of the legend in Figure 2. We have corrected it in the new version.
In Figure 3, the blot is clearly a hybrid cut and pasted from several different blots, since there is a comparison being made between VSV-G in different constructs how did the authors ensure that loadings on the gel etc were comparable between blots - the VP39 loading is not consistent between tracks shown. What is the doublet in tracks A -C?
We appreciate these important technical questions. The revised manuscript now includes uncropped blots (Supplementary Figure), normalized densitometry data (VSV-G/VP39), and new technical information. The doublet bands can be explained by potential VSV-G modifications or antibody cross-reactivity.
Figure 7 line 442 replace 'exposed' with 'transfected'? Were these representative images of those observed?
We appreciate the reviewer's suggestion. After careful consideration, we have replaced 'exposed' with 'transduced' as we agree this more accurately describes our experimental approach. Additionally, we confirm that the images presented are representative of all observations, and we have clarified this point in the revised figure caption.
Figure 8 line 448 Please explain more clearly what tissues are being viewed - a histological stained section may be useful to provide context of what is being look at? Are these muscle tissues?? Arrows to point to key features would also be useful. A little more detail in the results/materials and methods would be helpful too.
We appreciate the reviewer's helpful suggestion. The images in Figure 8 indeed show muscle tissue, and we acknowledge that this wasn't sufficiently clear in the original version. Following the reviewer's recommendation, we have added hematoxylin and eosin-stained sections to provide better histological context, including arrows to highlight key features in the images. Besides, we have expanded the descriptions in both Materials and Methods and Results sections to provide more detailed information about the tissues analyzed.
These changes should significantly improve the clarity and interpretability of our findings.

Round 2
Reviewer 1 Report
Comments and Suggestions for Authors
I have no other comments on this manuscript.
Author Response
Dear reviewers,
Thank you for your helpful feedback and for guiding us through the revision process. We sincerely appreciate the time and effort the reviewers have dedicated to improving our manuscript.
We have carefully considered your suggestion and are happy to confirm that we have updated the title to: "Early to Late VSV-G Expression in AcMNPV BV Enhances Transduction in Mammalian Cells but Does Not Affect Virion Yield in Insect Cells" (Note: I corrected "do not" → "does not" for grammatical consistency).
Please let us know if any further adjustments are needed. We are grateful for the opportunity to contribute to [Journal Name] and look forward to your final decision.
Best regards,
Mariano Belaich, PhD
Professor/Researcher, Department of Science and Technology
Universidad Nacional de Quilmes/CONICET
